# Pseudo-HFOs Elimination in iEEG Recordings Using a Robust Residual-based Dictionary Learning Framework

Behrang Fazli Besheli
*Department of Neurosurgery*
*Mayo Clinic*
Rochester, USA
fazlibesheli.behrang@mayo.edu

Zhiyi Sha
*Department of Neurology*
*University of Minnesota*
Minneapolis, USA
zysha@umn.edu

Amir Hossein Ayyoubi
*Department of Neurosurgery*
*Mayo Clinic*
Rochester, USA
ayyoubi.amirhossein@mayo.edu

Chandra Prakash Swamy
*Department of Neurosurgery*
*Mayo Clinic*
Rochester, USA
swamy.chandraprakash@mayo.edu

Tom Henry
*Department of Neurology*
*University of Minnesota*
Minneapolis, USA
trhenry@umn.edu

Gregory A. Worrell
*Department of Neurology*
*Mayo Clinic*
Rochester, USA
worrell.gregory@mayo.edu

Kai J. Miller
*Department of Neurosurgery*
*Mayo Clinic*
Rochester, USA
miller.kai@mayo.edu

Nuri Firat Ince
*Department of Neurosurgery and*
*Biomedical Engineering*
*Mayo Clinic*
Rochester, USA
ince.nuri@mayo.edu

*Abstract—* **High-frequency oscillations (HFOs) in intracranial EEG (iEEG) recordings are crucial for localizing the seizure onset zone (SOZ) in patients with focal refractory epilepsy. While HFOs are essential for clinical assessment, high-frequency artifacts may pass conventional HFO detectors, resulting in false-positive events that contaminate the HFO pool. The main goal of this study is to automatically detect and eliminate those false positive events in the pool of initially detected candidate HFOs. We analyzed one hour of iEEG data from fifteen patients with focal epilepsy using an attention-based cascaded residual dictionary learning framework, coupled with a random forest classifier. This data-driven method employed sparse robust representation using the Huber loss to eliminate artifacts and noises with non-neural origins that mimicked HFOs by evaluating the quality of event representation using a dictionary learned from real HFOs. Compared with visual assessments by three human experts, the proposed method achieved a 92.14% classification accuracy in distinguishing real HFOs from pseudo-HFOs. Additionally, in noisy iEEG data, our method improved HFO-based SOZ localization by 20% (p=6e-5), while in clean iEEG data, the improvement was 4% (p=3.3e-3). The learned dictionary successfully captured the morphology of raw HFOs in shallow layers, while it captured ripple and fast ripple components in deeper layers without human supervision. Our work shows that the proposed algorithm effectively detects pseudo-HFOs and improves the clinical value of HFOs in SOZ localization.**

*Keywords— Artifact Rejection, Cascaded Dictionary Learning, Drug-resistant Epilepsy, intracranial EEG, Focal Epilepsy HFO, Neurobiomarkers.*

## I. INTRODUCTION

Epilepsy is a medical condition characterized by the recurrence of unprovoked seizures due to abnormal electrical activity in the brain [1]. In cases where epilepsy is resistant to medication, iEEG-guided surgery is a technique employed to achieve seizure freedom [2]. The success of this procedure heavily depends on the accurate localization of epileptogenic brain tissue [3, 4]. Physicians typically rely on clinical judgment based on multiple seizures recorded in the epilepsy monitoring unit (EMU) to identify the seizure onset zone (SOZ). However, this method has several drawbacks, including the need for prolonged monitoring to capture enough clinical seizures, which makes the process resource-intensive, costly, and prone to human error. These limitations have led to the investigation of new prognostic biomarkers, such as high-frequency oscillation (HFO) [5, 6]. HFOs are transient oscillatory activities within the 80-600 Hz frequency range, typically lasting less than 100 ms [7]. Studies have shown that HFO rates are higher within the SOZ [8, 9], and resecting regions with high HFO rates can improve surgical outcomes [10, 11]. However, translating the use of HFOs into clinical practice is challenging due to several reasons. High-frequency artifacts due to the ringing effect introduced by filtering process can resemble HFOs [12, 13].

Additional challenges include physiological HFOs in healthy human brains [14, 15], sharp spike leakage into the HFO band [12], and HFO fluctuation over time [16]. While pseudo-HFOs can often be distinguished from real HFOs by their morphology, this process requires considerable time, labor, and expertise from clinicians [17]. Although numerous HFO detectors are available in the iEEG field [5, 18-20], they follow the same principle of identifying oscillations that stand out from the background and might capture pseudo-HFOs as real ones, making the interpretation of HFOs challenging.

We propose a data-driven pipeline that uses residual-based dictionary learning and a random forest (RF) classifier to distinguish real HFOs from pseudo-HFOs. By learning local waveforms from real HFOs, our method improves the delineation of HFO candidates to the SOZ and reduces false positives caused by artifacts.

This study was supported by the National Institutes of Health's BRAIN Initiative under award number UH3NS117944 and grant R01NS112497 from the National Institute of Neurological Disorders and Stroke. B.F.B. was supported by the Sundt fellowship of the Mayo Clinic Neurosurgery Dept.

## II. MATERIALS & METHODS

### A. Dataset

The iEEG recordings were obtained from 15 patients with drug-resistant focal epilepsy from the University of Minnesota Medical Center (MN, USA). A brief 30-minute interictal noisy iEEG segment and a 30-minute-long noise-free iEEG segment are visually identified and extracted from the first day of the prolonged monitoring. All recordings were obtained at a sampling frequency of 2 kHz. The entire HFO analysis pipeline was blinded to the relevant clinical data such as SOZ, surgical procedure, and outcome of the surgery. Furthermore, the iEEG data of the entire cohort of subjects went through the same offline analysis without any preprocessing or channel selection. The research protocol was approved by the institutional review board at the Mayo Clinic and the University of Minnesota. Epileptologists reviewed the iEEG recordings in each subject and provided the required annotations, including the SOZ channels. Further details regarding the data used in this study can be found in Table 1.

### B. Algorithm Overview

This work introduces a cascaded residual-based dictionary learning framework, where dictionaries are learned from HFO events. Using these learned dictionaries and a robust regression representation, we effectively distinguished between true HFOs and high-frequency artifacts that resemble HFOs after filtering in the 80-600 Hz band. Fig. 1 illustrates the two-step process: first, a dual-band amplitude threshold detector identifies candidate HFOs from the raw iEEG recording. Then, these candidates are classified as real and pseudo-HFOs by the RF classification framework using features extracted from the waveform patterns present in HFOs. The automatic classification pipeline utilizes features that capture how well the system can represent true HFOs and how poorly it represents pseudo-HFOs. This is achieved using a learning technique based on attention-based cascaded

TABLE I.    PATIENT DEMOGRAPHICS

| Subject | Implanted Electrodes | Contacts | Treatment | SOZ | Engel Class Outcome |
|---|---|---|---|---|---|
| **P1** | 3 depths (LAH, RA, RAH) and 1 strip (LAS) | 28 | Right anterior temporal lobectomy | RAH1-2 | II. A at 5 years |
| **P2** | 6 depths (LA, LAH, LPH. RA, RAH, RPH) and 1 strip (LftAT) | 52 | Left anterior temporal lobectomy | LA1-2 | I. B at 9 years |
| **P3** | 6 depths (LA, LAH, LPH. RA, RAH, RPH) and 1 strip (LftPF) | 56 | No resection or other surgical therapy. | LA1-3, LAH1-2, LPH1-3 | N. A |
| **P4** | 6×8 grid, 6 strips (SupF, MidF, InfF, Par, AST, PST) | 72 | Right anterior temporal lobectomy | AST1, 4, PST1, 4 | I.A. at 2 years. |
| **P5** | 8×8 grid, 1 strip (MedF) | 68 | Right superior temporal gyrus resection. | G1-4, 9-12, 17-18, 25-26, 33-34, 41-42 | I.C., at 6 years. |
| **P6** | 13 strips (LAT, LMT, LPT, LOF, LAP, LPP, RAT, RMT, RPT, ROF, RAP, RPP, L/RIH) | 56 | Right anterior temporal lobectomy | LAT1-4, LMT1-4, RPT1-4, RMT1 | I.A. at 1 year. |
| **P7** | 7 depths (RFP, RMF, RAC, ROF, RAH, RPH, RPT) | 60 | Right anterior temporal lobectomy | RAH1-2, RPH1-2, RPT1-2 | II.A. at 4 years. |
| **P8** | 8 depths (RAH, RH, RPH, ROF, RI, RAC, LAH, LPH) | 108 | Right anterior temporal lobectomy | RAH1-5, RPH1-4, RH1-5 | II.A. at 1 year. |
| **P9** | 8 depths (LA, LAH, LPH, LOF, LAC, LPC, ASP, LC) | 90 | Left amygdalohippocampal laser thermablation | LAH1-4, LPH1-4 | I.A., at 5 months. |
| **P10** | 4 strips (LAT, LMT, LPT, LFPT), 10 depths (LAMG, LAH, LPH, RAMG, RAH, RPH, LOF, LDLPFC, ROF, RDLPFC) | 128 | RNS, with bilateral amygdalohippocampal DEs | LAMG1-3, LAH1-3, LPH1-5 | I.A. at 5 months |
| **P11** | 10 depths (RAH, RPH, ROI, RAC, RAS, LAH, LPH, LOF, LAC, LAS) | 144 | Right anterior temporal lobectomy | RAH1-4, RPH1-4 | II.A. at 1 month. |
| **P12** | 10 depths (RA, RAH, RPH, RF, RAC, RPC, RAL, RSL, RIL, RPL) | 108 | Right amygdalohippocampal laser thermablation | RAH1-4, RPH1-4, RIL6-8, | I.D. at 5 months. |
| **P13** | 9 depths (LMA, LMP, LMM, LFA, LFP, LSM LSL) | 68 | RNS, with bilateral temporal depth electrodes, in hippocampi | LMA, LMP, LMM, LML, LSM, LSL | II.A. at 14 months. |
| **P14** | 4x6 grid, 4 strips (RMM, RMS, LCSM, LCSL), and 6 depths (RMFA, RMFB, RMFC, LMFA, LMFB, LMFC) | 86 | RNS at sites of ICEEG recoding contacts RMM 2-4 and RMS 2-4. | G13, 16, 19-22, RMM2-4, RMS 2-4 | IV.A. at 17 months. |
| **P15** | 15 depths (RA, RAH, RPH, ROF, RAC, RAMC, RI, LA, LAH, LPH, LOF, LAC, LAMC, LI, RPL) | 128 | RNS with bilateral temporal depth electrodes, in hippocampi | LAH1-4, LPH1-4, RAH1-4, RPH1-4 | III.A. at 2 months. |

**Abbreviation: LA/LAMG:** Left Amygdala, **LAH:** Left anterior hippocampus, **LPH:** Left posterior hippocampus, **RH:** right mid hippocampus, **RA/RAMG:** Right Amygdala, **RAH:** Right anterior hippocampus, **RPH:** Right posterior hippocampus, **SupF:** Superior frontal, **MidF:** Middle frontal, **InfF:** Inferior Frontal, **Par:** Parietal, **AST:** Anterior subtemporal, **PST:** Posterior subtemporal, **MedF:** medial frontal gyrus, **LAT:** Left anterior temporal, **LMT:** Left middle temporal, **LPT:** Left posterior temporal, **LOF:** Left orbital frontal, **LAP:** Left anterior parietal, **LPP:** Left posterior parietal, **RAT:** Right anterior temporal, **RMT:** Right middle temporal, **RPT:** Right posterior temporal, **ROF:** Right orbital frontal, **RAP:** Right anterior parietal, **RPP:** Right posterior parietal, **L/RIH:** Left and Right interhemispheric strip (double sided), **RFP:** right middle frontal gyrus, **RMF:** right middle frontal gyrus, **RAC:** Right anterior cingulate, **RI:** Right Insula, **LAC:** Left anterior cingulate, **LPC:** Left posterior cingulate, **ASP:** Left anterior-superior precuneus, b Left anterior cuneus, **LFPT:** Left temporal neocortex, **L/RDLPFC:** Left/Right  cingulate, **ROI:** right orbital frontal, **RAS:** right anterior subfrontal, **LAS:** Left anterior subfrontal, **RF:** Right superior frontal gyrus, **RPC:** Right posterior cingulate, **RAL:** Right anterior lesion, **RSL:** Right superior lesion, **RIL:** Right inferior lesion, **RPL:** Right posterior lesion, **LMA:** Left motor anterior, **LMP:** Left motor posterior, **LMM:** Left motor medial, **LML:** Left motor lateral, **LFA:** Left frontal anterior, **LFM:** Left frontal middle, **LFP:** Left frontal posterior, **LSM:** Left sensory medial, **LSL:** Left sensory lateral, **RAMC:** Right ant. to mid. cingulate, **LAMC:** Left ant. to mid. cingulate, **RMM:** Right medial motor, **RMS:** Right medial sensory, **LCSM:** Left cortical strip medial, **LCSL:** Left cortical strip lateral, **RMFA/B/C:** Right malformation, **RNS:** Responsive neural stimulation

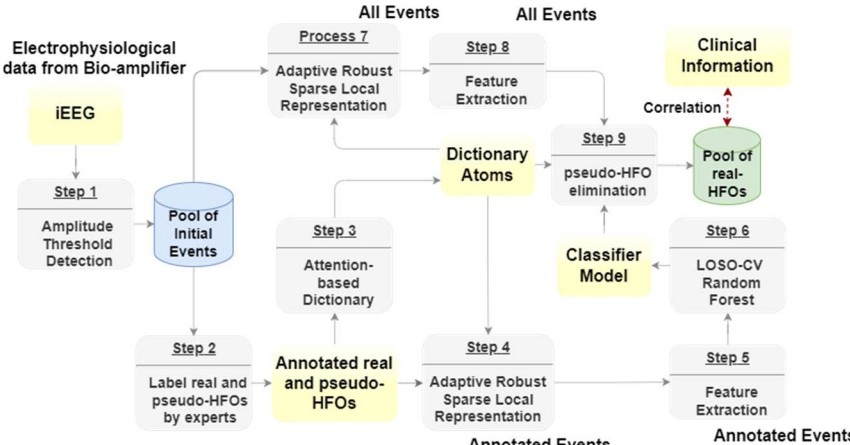

Fig. 1. Schematic of the proposed method. The processing pipeline begins with applying an amplitude threshold detector to the raw iEEG data, creating an initial pool of events. Three experts annotate a portion of this pool to develop the dictionary and RF model. The model is then applied to the entire initial pool using leave-one-subject-out cross-validation to eliminate pseudo-HFOs from the initial pool. Finally, the spatial distribution of remaining HFOs is compared with the clinically defined SOZ.

dictionary learning. Finally, the spatial distribution of the classified HFOs is compared with the clinically defined SOZ to evaluate the improvement in SOZ delineation with and without pseudo-HFO removal in both noisy and clear iEEG data.

### C. Amplitude Threshold Detection

A previously published detector was utilized to generate an initial pool of candidate HFO events [13]. This detector is a dual-band amplitude threshold detector that identifies events in the ripple (R) and fast ripple (FR) bands separately and merges them if they occur close together (within 30 milliseconds). Each candidate event includes a window of data (128 milliseconds before and after the peak high-frequency component) to capture the surrounding activity. This window corresponds to 512 data points at the sampling rate of 2 kHz. The detector ensures a minimum number of 6 threshold crossings in either positive or negative cycle to increase the confidence in the event as a real HFO.

### D. HFO and pseudo-HFO Annotation

To train the classifier, a subset of the detected events needed to be labeled as real or pseudo-HFOs. Three experts independently annotated up to 400 events per subject, aiming for a balanced label of real and pseudo-HFOs (if sufficient real and pseudo-HFOs were detected). A custom graphical user interface (GUI) facilitated the annotation process by visualizing relevant information, including the multi-channel iEEG data surrounding the events, the time-frequency map, and the spatio-temporal distribution of the detected events. This comprehensive view of the data helped the annotators make informed decisions and minimize labeling errors. Finally, using majority voting, the labeled dataset is created to train the classifier.

### E. Attention-based Dictionary Learning

Fig. 2 illustrates the overall diagram of the proposed dictionary learning framework. This method addresses the challenge of capturing the diverse characteristics of HFOs due to their varying spectral levels. The annotated HFO (distinct from the real HFOs used for cross-validation), denoted as $[Ev]_{n \times 512}$ ($n$: total number of events) are initially processed through a shallow layer (Layer I), where they are

buffered into overlapping 64 ms segments, resulting in $[Y^1]_{n1 \times 128}$ ($n_1$: total number of buffered events). The k-SVD dictionary learning algorithm [21] is then applied to $Y^1$, capturing the most important patterns within these segments and forming the dictionary $D^1$. The k-SVD algorithm seeks to solve the following optimization problem:

$$D^l, X = \underset{D^l, X}{argmin} \left\| Y^l - D^l . X \right\|_2 \qquad l = 1, .., 4 \qquad (1)$$

where $D^l$ is the dictionary at layer $l$, and $X$ represents the coefficients of representation.

Due to the energy difference and the $1/f$ characteristic of

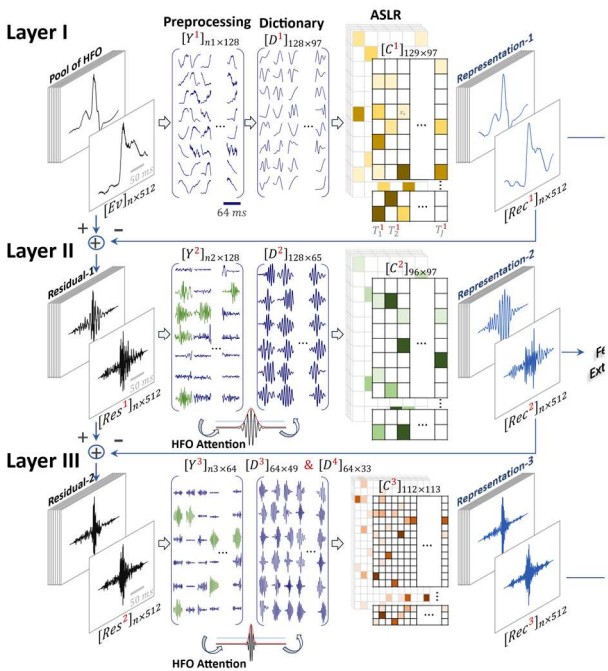

Fig. 2. Diagram of dictionary learning and robust representation strategy. The dictionary is learned on annotated real-HFOs, using distinct and non-overlapping events with those used for cross-validation. The first layer's input consists of annotated HFOs, while deeper layers learn atoms from the residuals of the previous layer using attention-based dictionary learning. The representation of all events is computed using an adaptive sparse local representation with OMP/Huber regression applied on a moving window over a buffered event.

iEEG, and to effectively capture higher frequency components, particularly oscillatory transients above 80 Hz, we introduced an attention-based residual cascaded framework. Here, the initial reconstruction of the events $[Ev]_{n \times 512}$ using the first dictionary $D^1$ is subtracted from the original events, resulting in a residual signal $[Res^1]_{n \times 512}$:

$$Res^1 = Ev - Rec^1 \qquad (2)$$

where $[Rec^1]_{n \times 512}$ is the reconstructed event using $D^1$. This residual signal is then fed into the next layer (Layer II). The same process, including buffering $Res^1$ to $[Y^2]_{n2 \times 128}$ ($n_2$: total number of buffered events passed HFO criteria) and k-SVD dictionary learning is applied. However, at this stage, to focus on transient oscillatory HFO components, an additional step called HFO attention is applied between buffering and dictionary learning. In this scheme, an envelope-based HFO-component detector is applied to the $Y^2$ samples. Only segments that pass the HFO attention criteria—specifically, those that stand out from the background and surpass the detector's threshold at least six times—are used for dictionary learning. Consequently, in deeper layers (Layer > 1), we exclusively learn higher frequency oscillatory components. The dictionary $D^2$ is then learned, and using $D^2$, we represent all $Res^1$ events. Following the same procedure, the represented $[Rec^2]_{n \times 512}$ is subtracted from $Res^1$, yielding $[Res^2]_{n \times 512}$ as the input to the subsequent layer. This process of buffering, HFO attention, and k-SVD dictionary learning is iteratively applied, resulting in the learning of subsequent dictionary layers. Similarly, in these deep layers, HFO attention is utilized to discard segments without oscillatory high-frequency components distinguishable from background activity. Overall, we trained dictionaries consisting of 96, 64, 48, and 32 atoms and included a normalized DC component into each layer of the dictionary to account for local DC levels of waveforms. The first two layers contained local atoms with 128 sample sizes, while the last two layers had 64 sample sizes. ($[D^1]_{97 \times 128}$, $[D^2]_{65 \times 128}$, $[D^3]_{49 \times 64}$, and $[D^4]_{33 \times 64}$).

*F. Representation of Events Using Local Dictionaries*

The learned dictionary was employed in each layer to represent all initially detected events via adaptive sparse local representation (ASLR) algorithm [22]. To achieve this, buffered segments are first represented using a sparse combination of the learned dictionary. Thus, the objective of this step is to represent all buffered segments $Y^l$ using $D^l$ at each layer. We merged dictionary layer-3 and 4 into one level to leverage the redundancy in learned FR band atoms. Therefore, the representation process involved 3 stages in a cascaded fashion. Using these dictionaries, we compared different approaches to represent the local waveforms using these dictionaries. One common method is the Orthogonal Matching Pursuit (OMP) [23]. The OMP representation of a segment $Y^l$ using the dictionary $D^l$ is formulated as:

$$X_{OMP} = \underset{X}{argmin} \left\| Y^l - D^l.X \right\|_2 \; subject\; to \; \|X\|_0 \leq T \qquad (3)$$

where $X$ represents the sparse coefficients, and $T$ is the sparsity level. Alternatively, we can use a robust representation method that employs the Huber loss function

for optimization. The Huber loss function is a smooth $L_1$ loss making the error term less sensitive to outliers compared to OMP. The robust representation using Huber loss regression is formulated as:

$$X_{Huber} = \begin{cases} \frac{1}{2}\left\|Y^l - D^l.X\right\|_2^2, & |a| \leq \delta \\ 2\delta\left|Y^l - D^l.X\right| - \delta, & |a| > \delta \end{cases} \qquad (4)$$

where $\delta$ is the threshold parameter at which the loss function transitions from a quadratic to a linear regime.

Robust representation minimizes the influence of outliers (sharp artifacts), ensuring an accurate representation of real HFOs (Fig. 3, top panel). In contrast, delta-shaped artifacts in pseudo-HFOs are not mitigated, leading to inefficient representation of pseudo-HFOs (Fig. 3 bottom panel). Using the initial buffered data, the representation in each stage acts like a moving window across the entire buffered data (Fig. 4). This process can be mathematically expressed as:

$$\sum_{j=1}^{J} (\underset{X^{(k)}}{min}\left\|Y^l - D^l.X^{(k)}\right\|_2^2) * \delta(t - T_j^l.d^l) \qquad (5)$$

Here, $T_j^l$ represents the time index of the $jth$ segment in layer $l$, and $d^l$ represents the corresponding dictionary atom.

This work introduces a robust regression attention-based cascaded residual

*G. Feature Extraction*

Real-HFO events have been shown to exhibit structured patterns [13]. Since these local dictionaries were learned from these real-HFOs, they are likely to be more effective at capturing the characteristics of these events compared to pseudo-HFOs. Therefore, the quality of representation can be used to distinguish between real and pseudo-HFOs. The following features are extracted to quantify the quality of representation:

**Global Approximation Error:** This metric assesses the overall representation quality of an event in each layer $l$. It is defined as:

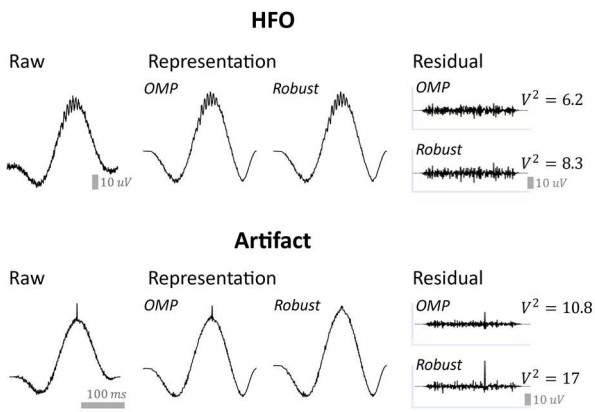

Fig. 3. Robust vs. OMP Representation. It compares the robust representation using the Huber loss function and OMP representation. In the top panel, a raw HFO is represented using both OMP and robust representations, with comparable results due to the oscillatory nature of HFO components. In the bottom panel, depicting a sharp delta-shaped artifact, the OMP representation begins to reconstruct the delta activity. In contrast, the robust representation identifies the delta-shaped activity as an outlier, leaving it untouched.

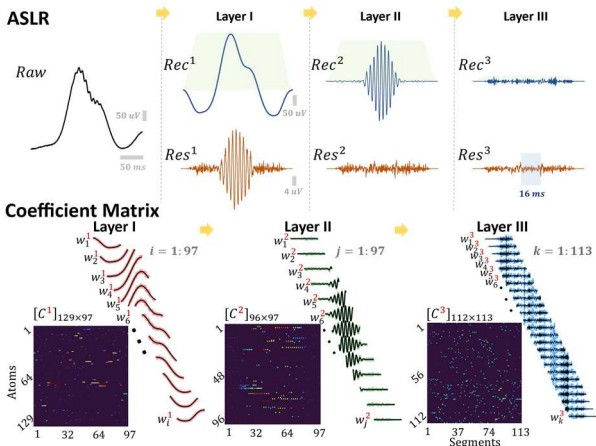

Fig. 4. Adaptive sparse local representation. The top panel illustrates the hierarchical representation of the raw HFO, along with the corresponding residuals at each layer. The bottom panel shows local waveforms (in color) and the corresponding reconstructed waveforms (in black) for buffered segments across layers. Additionally, the coefficient matrix is displayed, with the x-axis representing time segments and the y-axis indicating the dictionary atoms used, along with their associated coefficients.

$$\varepsilon^l = \frac{\|Res^l\|_2}{\|Res^l + Rec^l\|_2}, l = 1,2,3. \quad (6)$$

**Variability Factor:** This feature determines how smoothly the waveform is represented in each layer. It is calculated as:

$$V^l = \frac{(max(Res^l) - min(Res^l))}{std(Res^l)}, l = 0,1,2 \quad (7)$$

where $Res^0$ is equal $Ev$.

**Maximum Coefficient of Representation:** After the representation of all segments, we calculate the coefficient matrix $C^l$ for the reconstruction phase at layer $l$. The maximum coefficient within this matrix indicates the degree of similarity between the event at each layer and the learned dictionary:

$$CM^l = max([C^l]), l = 1,2. \quad (8)$$

**Maximum Improvement in Approximation Error:** To investigate the impact of increasing the sparsity level or the number of atoms used to represent each segment, we computed the local approximation error matrix for varying sparsity levels. Additionally, we analyzed the degree of improvement in the quality of representation obtained by increasing the number of atoms employed for the representation. Therefore, the maximum improvement in approximation error achievable by increasing the sparsity level is defined as $DM^l, l = 1,2$.

**Maximum Eigen Value:** The distortion of power line or harmonic interference in iEEG recordings might create artificial oscillations that mimic HFOs. To address these pseudo-HFOs, we introduced a feature based on the repetition of specific atoms in the coefficient matrix. By examining the maximum eigenvalue of this matrix, we can effectively identify and exclude anomalous repetitive non-neural patterns:

$$EVT^1 = max(eig([C^1]^T \times [C^1])). \quad (9)$$

**Overall central error:** The overall representation of an event can be calculated by summing up the representation in each layer $l$. Therefore, we expect to observe a full representation in the case of real-HFO and overall error reaching the amplifier background noise. Whereas, in the case of pseudo-HFO, we expect to have a higher approximation around the center of events where high-frequency components exist:

$$CE = \left\| Res^3_{(\frac{N}{2}-16, \frac{N}{2}+16)} \right\|_2 \quad (10)$$

In addition to these features, the range of each event was included to the extracted features for generalizability:

$$rng = max(Ev) - min(Ev) \quad (11)$$

The AUC and ROC values of extracted features using OMP, and robust representation are summarized in Fig. S1. Fig. 5 provides comprehensive examples of HFOs and

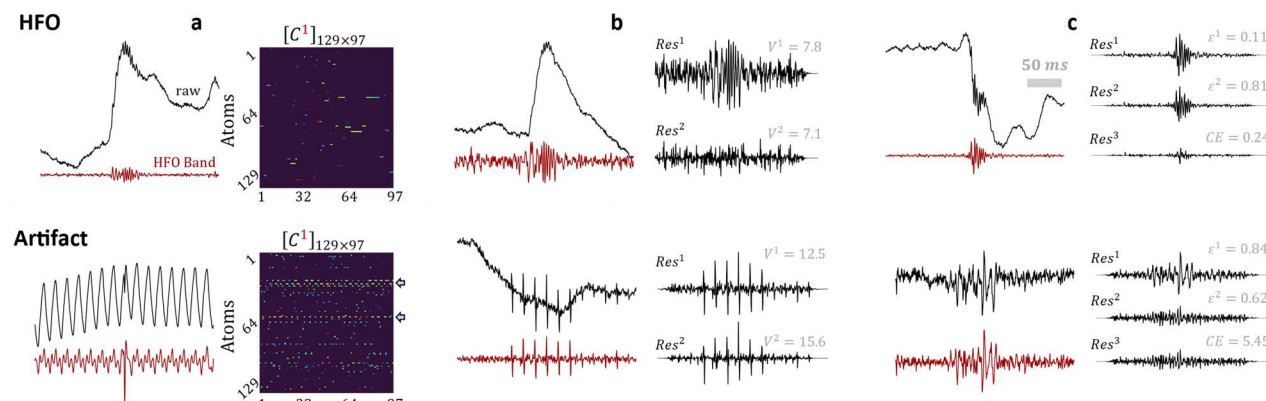

Fig. 5. Real vs. pseudo-HFOs. This figure presents three examples of HFOs and pseudo-HFOs. (a) Shows one HFO and pseudo-HFO example to visualize the coefficient matrix at layer 1, quantifying the importance of the first eigenvalue of the coefficient matrix in identifying pseudo-HFOs with repetitive patterns. Marked by blue arrows, artifacts with repetitive patterns exhibit high eigenvalues due to the presence of atoms repeating during the representation phase, which is not observed in real HFOs. (b) Displays an example of HFO and pseudo-HFO with sharp delta-shaped activity. While HFOs are accurately represented, the delta-shaped component of pseudo-HFOs remains untouched due to the absence of dictionary atoms to represent such waveforms, resulting in a high v-factor in the second and third layers of representation. (c) Demonstrates another example of HFO and pseudo-HFO, where the approximation error for HFOs in all layers is low. In contrast, pseudo-HFOs with corrupted channels or thick background activity exhibit high global approximation errors during the representation process across different layers.

pseudo-HFOs to illustrate their characteristics. In Fig. 5a, an instance of an HFO and a pseudo-HFO contaminated by line noise is depicted, showing a repetition of a few atoms compared with real-HFOs in the coefficient matrix. Fig. 5b visualizes the concept of a high V-factor associated with artifacts exhibiting delta-shaped activities, while Fig. 5c demonstrates pseudo-HFOs characterized by random fluctuations or corrupted channels.

## H. Classification of events to real and pseudo-HFOs

We used an RF model and performed leave-one-subject-out cross-validation (LOSO-CV) to ensure the generalizability of the proposed method across different subjects. To validate the effectiveness of the proposed method, we conducted following comparisons: We compared the classification performance with an analytical Gabor dictionary with global representation. This demonstrates the effectiveness of using a local dictionary instead of a global dictionary. Moreover, we included a local analytical dictionary based on the discrete cosine transform (DCT) to compare the efficacy of the learned dictionary and whether this data-driven, cascaded dictionary learning approach outperforms a pre-defined codebook. We also compared OMP and Huber regression for the representation phase. This comparison emphasizes the importance of robust representation using Huber regression, which minimizes the influence of outliers present in pseudo-HFOs and leads to improved classification accuracy.

## III. RESULTS

### A. Learned Atoms

In Fig. 6, an example set of learned dictionaries is presented using the proposed method. As shown, the shallow layers capture lower frequencies, with the first layer focusing on frequencies below 80 Hz, i.e., the morphology of events.

The second layer captures frequencies within the 80-250 Hz range, corresponding to the R band. The third and fourth layers extend into the FR band, capturing frequencies beyond 250 Hz. Remarkably, despite the absence of human intervention, three distinct and well-defined patterns emerged across the layers of the dictionary. These patterns closely align with the conventional definitions of R and FR bands, demonstrating a clear energy distinction between these frequencies which creates different clusters or groups of atoms in low, R, and FR bands.

### B. Classification Results for Annotated HFOs and Pseudo-HFOs

After learning the dictionary, all annotated events went through a representation process. The sparsity level used for representation in each layer is explained in Fig. S2. The relevant features were extracted, and all initially detected events were classified as real and pseudo HFOs. The classification performance using LOSO-CV is summarized in Fig. 7. These results were compared with the RF-OMP method [13], which utilizes a global overcomplete predefined dictionary, achieving an accuracy of 90.79%. Additionally, a local overcomplete learned dictionary based on proposed dictionary learning and ASLR sparse coding was employed. For the OMP-based ASLR representation, the accuracy was 91.17%, and for the robust Huber-based regression representation, it increased to 92.14%. We used a pre-defined windowed DCT dictionary with the same frequency range for each layer as learned in dictionary learning framework. This approach resulted in an accuracy of 88.43%, highlighting the effectiveness of data-driven and dictionary learning approaches in distinguishing between real and pseudo-HFOs. The comparison underscores the superiority of using a local dictionary over a global one. The local dictionary could better capture the local characteristics and variations of real-HFOs, leading to higher accuracy in distinguishing between real and

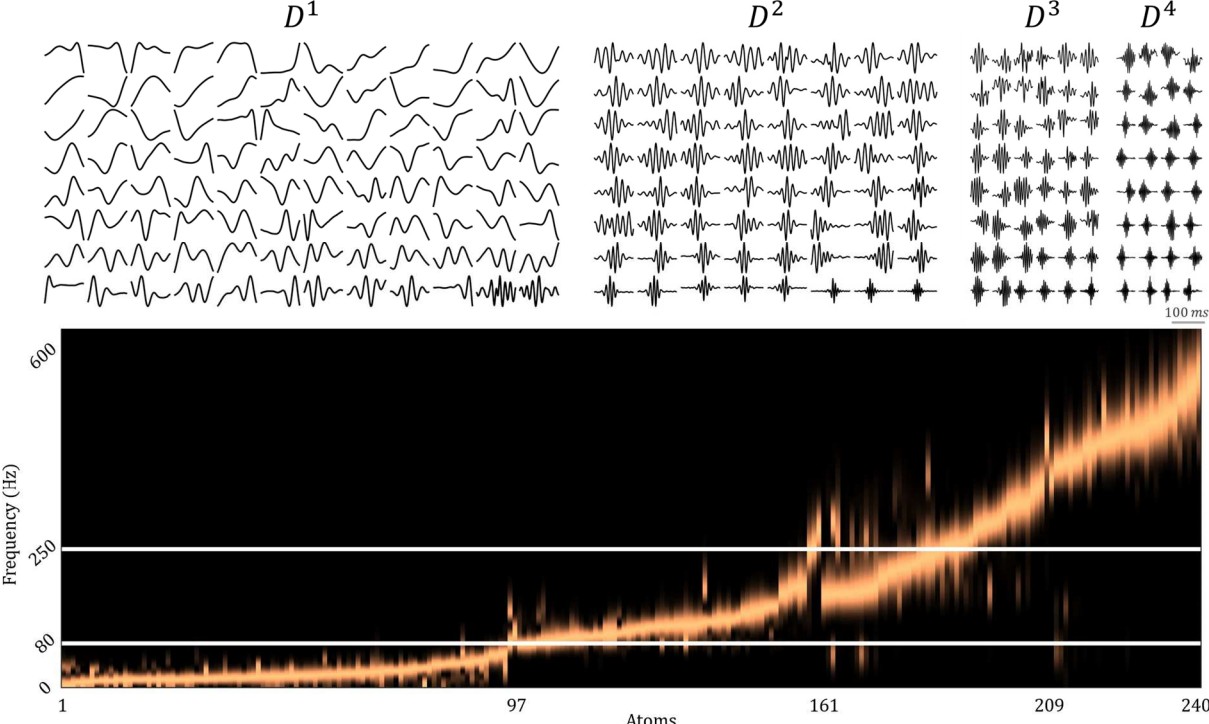

Fig. 6. Example of Learned Dictionaries. The top panel visualizes the learned dictionary within each layer. Shallow layers learn atoms with frequencies below 80 Hz, while deeper layers begin to learn HFO components in the ripple band (layer 2) and fast ripple band (layers 3 and 4). The bottom panel visualizes the frequency of these learned atoms, which aligns with the conventional definition of R and FR components.

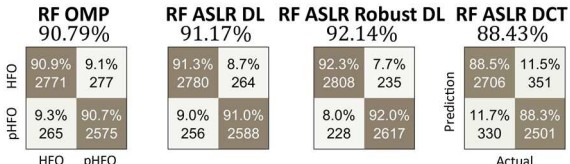

| RF OMP 90.79% | | RF ASLR DL 91.17% | | RF ASLR Robust DL 92.14% | | RF ASLR DCT 88.43% | |
|---|---|---|---|---|---|---|---|
| 90.9% 2771 | 9.1% 277 | 91.3% 2780 | 8.7% 264 | 92.3% 2808 | 7.7% 235 | 88.5% 2706 | 11.5% 351 |
| 9.3% 265 | 90.7% 2575 | 9.0% 256 | 91.0% 2588 | 8.0% 228 | 92.0% 2617 | 11.7% 330 | 88.3% 2501 |

Fig. 7. Comparison of real and pseudo-HFO (pHFO) classification methods: the sparse representation with a global predefined codebook (RF-OMP [13]) achieved an accuracy of 90.79%. In comparison, the proposed dictionary learning method with OMP representation and robust representation achieved accuracies of 91.17% (RF ASLR DL) and 92.14% (RF ASLR Robust DL), respectively. When the learned dictionary was replaced with predefined DCT components, the accuracy was 88.43% (RF ASLR DCT) for annotated events.

pseudo-HFOs. Additional analysis on the precision, recall, and accuracy of different methods across each subject is provided in Fig. S3.

### C. Delineation to Clinically Defined SOZ

To evaluate this pseudo-HFO elimination method in a real-world scenario, we applied the entire framework, including initial detection and pseudo-HFO elimination, to raw iEEG data. The analysis utilized 30 minutes of noisy and 30 minutes of clean data selected through visual observation of the raw iEEG by electroencephalographers. We then assessed the spatial distribution of HFO subcategories, i.e., Rs and FRs (Fig. 8 right panel). The cosine distance similarity between the spatial distributions of the R and FR groups in the clean and noisy data was calculated as $d=cos\theta$ in which θ is the angle between the spatial distribution of R/FR rate for noisy and clean data. Initially, using conventional amplitude threshold detector, a significant misalignment between the distributions of detected Rs and FRs was observed due to the presence of artifacts (R: 0.93 FR: 0.87). However, after applying the

denoising method, the spatial alignment of R and FR groups between noisy and clean data improved significantly, achieving higher similarity (R: 0.95, FR: 0.98). This improvement indicates that ripples are less susceptible to artifact corruption. Moreover, the distribution of FRs is more stable post-denoising (0.98 across noisy and clean segments after denoising).

In terms of clinically defined SOZ, we assessed the accuracy of SOZ localization based on the spatial distribution of HFOs. We calculated the ratio of HFOs (and their subgroups) within SOZ channels before and after pseudo-HFO elimination (refer to Fig. 8 left panel). The agreement bubble plot, displaying all HFOs before and after the denoising method, shows that in clean data, all data points, regardless of their overall rate, align along the diagonal of the agreement plot. This indicates minimal improvement in SOZ localization after applying the denoising method, as the segment was already clean without much spurious events (initial: 65% vs. denoised: 69%, p=3.3e-3). In noisy data, all subject data points are clustered in the upper left diagonal part of the agreement plot. This implies a consistent improvement in SOZ localization for all subjects after denoising (initial: 42%, denoised: 62, p=6e-5). The initial vs. denoised box plot for Rs and FRs shows that Rs are less affected by artifacts. However, there was a notable improvement in SOZ localization for FRs in both clean and noisy data (clean data: 75% to 84%, p=9.8e-3; noisy data: 48% to 73%, p=3.9e-3). This highlights the effectiveness of the denoising or pseudo-HFO elimination in enhancing SOZ localization, particularly in artifact-laden recordings.

## IV. SUMMARY & DISCUSSION

This work presents an HFO detection framework using an attention-based cascaded residual dictionary learning approach, which does not require artifact-free data. This

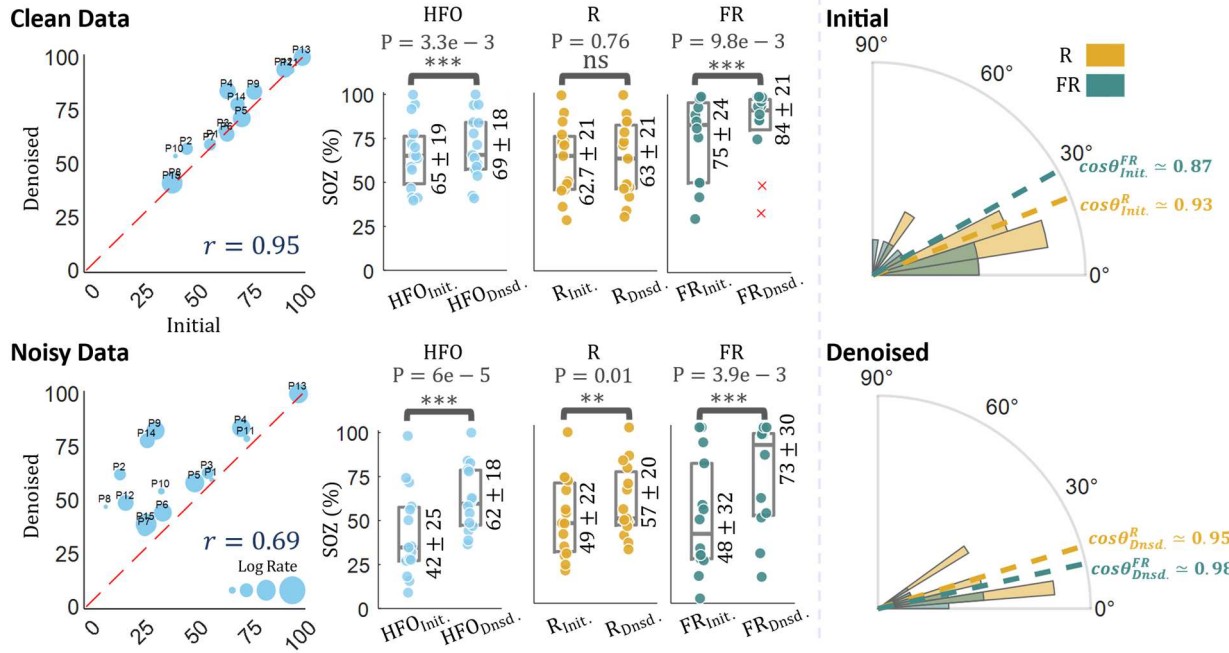

Fig. 8. Correlation of HFO analysis and clinical data. The top left panel displays the agreement plot before (x-axis) and after (y-axis) pseudo-HFO elimination in clean data. Similarly, the bottom left figure visualizes this comparison for noisy segments. The boxplot of HFO SOZ accuracy for HFO and their subcategories (R and FR) is shown before and after pseudo-HFO elimination in both clean and noisy data (ns. indicates p>0.05). The right panel illustrates the cosine similarity between HFO subgroups in clean and noisy data before and after pseudo-HFO elimination across all subjects. While the similarity was low before pseudo-HFO elimination, after pseudo-HFO elimination, we observed similar R and FR distributions in noisy and clean data.

approach enhances resource efficiency, reduces healthcare costs, and shortens the time needed for SOZ identification, improving the precision of epilepsy care. This two-stage framework addresses the challenge of pseudo-HFOs in iEEG data, which can hinder accurate SOZ localization using HFO. The core of this method lies in the learned dictionaries: the first layer of the dictionary captures the morphology of HFOs, while subsequent layers identify HFO components through attention-based residual learning. Our data-driven method outperforms pre-defined analytical dictionaries. The robust regression used in our method is superior to the traditional OMP method because it is less biased towards outliers, such as delta-shaped artifacts.

Results demonstrate that the quality of representation can be a distinctive feature between real and those pseudo-ones with non-neural origin. This indicates that HFOs possess an inherent structure validated by the learned dictionaries. Additionally, the evaluation of HFO distribution in prolonged EMU has gained significant interest [16]. Our study highlights the crucial role of data quality in this analysis. We observed that including segments with artifacts can skew HFO analysis. This emphasizes the importance of eliminating pseudo-HFOs, particularly in noisy data, before interpreting the spatial distribution of HFOs. Therefore, future studies exploring HFO fluctuations within prolonged data should carefully consider the quality of the iEEG data to ensure an accurate interpretation of the results. Moreover, we showed FRs are more susceptible to corruption by high-frequency artifacts. In noisy data, the delineation of FRs to SOZ is lower than Rs but improved dramatically after pseudo-HFO elimination. additionally, further research is needed to evaluate the ability of a learned dictionary to capture variability within subjects and identify potential groups of atoms more specific to the SOZ.

The proposed framework involves multiple stages, which increase its computational complexity (Table S1). To enhance its clinical utility, future research could explore the use of a convolutional process or shift-invariant type of representation to speed up the algorithm.

Finally, while the proposed method demonstrates flexibility in identifying pseudo-HFOs from the pool of candidates, this study is based on a relatively small dataset of 15 patients. Further validation on larger and more diverse datasets is necessary to establish the robustness and generalizability of the method.

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
