# OpenReview forum: "Pseudo-HFOs Elimination in iEEG Recordings Using a Robust Residual-based Dictionary Learning Framework"
_IEEE.org/EMBS/BHI/2024/Conference — IEEE BHI'24_

### Official Review · Reviewer_bwuB · 2024-08-08
**A technically solid paper with good clarifications**

**Overall Rating:** 7
**Confidence:** 3

**Other Quality Metrics:**

(a) Clarity of writing - great;
(b) Clinical Significance - good;
(c) Methodological Novelty - good;
(d) Experiments and Results - good

**Questions For The Authors:**

My concerns are mostly summarized as the weaknesses in the review comment.

**Strengths:**

- The paper addresses a challenge in epilepsy surgery, which is the accurate identification of HFOs for SOZ localization. The proposed method is designed to distinguish true HFOs from pseudo-HFOs, which can impact surgical outcomes.
- The authors employ a combination of dictionary learning, sparse representation, and machine learning techniques, which demonstrates a sound understanding of signal processing and pattern recognition principles.
- The study includes a comprehensive evaluation of the proposed method on iEEG data from 15 patients with focal epilepsy. The use of both noisy and clean data segments and the comparison with visual assessments by experts provide valuable insights into the method's effectiveness and generalizability.
- The results demonstrate a significant improvement in HFO-based SOZ localization, particularly in noisy iEEG data.

**Summary Of The Paper:**

This paper presents a novel approach to address the challenge of distinguishing true high-frequency oscillations (HFOs) from artifact-induced pseudo-HFOs in intracranial EEG (iEEG) recordings. The authors propose a two-stage framework that combines a conventional amplitude threshold detector for initial HFO detection with a subsequent refinement process based on residual dictionary learning and a random forest classifier. The method leverages the inherent structure of real HFOs by learning dictionaries from annotated examples and using them to represent and classify detected events. The study demonstrates the effectiveness of the proposed approach in improving the accuracy of HFO-based seizure onset zone (SOZ) localization, particularly in noisy iEEG data. The results highlight the potential of the method to enhance the clinical value of HFOs as biomarkers in epilepsy surgery.

**Weaknesses:**

- The study is based on a relatively small dataset of 15 patients. While the results are promising, further validation on larger and more diverse datasets is necessary to establish the robustness and generalizability of the method.
- The proposed framework involves multiple stages and techniques, which might increase its computational complexity and implementation challenges. The authors could discuss potential strategies for optimizing the algorithm and making it more accessible for clinical use.
- While the method demonstrates high accuracy, the interpretability of the learned dictionaries and the features used for classification could be further explored. Providing insights into the specific patterns captured by the dictionaries and their relationship could enhance the clinical understanding and acceptance of the method.

---

### Official Review · Reviewer_jzEh · 2024-08-10
**Good evaluations with a methodological novelty**

**Overall Rating:** 7
**Confidence:** 3

**Other Quality Metrics:**

Clarity of writing: Great
clinical significance: Great
methodological novelty: Great
experiment and results: Good

**Questions For The Authors:**

- Why is only 30 minutes of data selected per patient? Can you comment on the run-time of the algorithm, if that was the major concern for not using larger segments?

- Is it possible to include the dictionary learning within the cross-validation loop and report results? Such evaluation will be more unbiased.

- The residual learning makes sense. But how would it compare with learning parallel dictionaries at different frequency decompositions?

**Strengths:**

The paper is well-written and easy to follow. The evaluation is based on a real-world dataset. There is some methodological novelty in the proposed dictionary learning in a residual cascaded network, which in the benchmarks outperforms other dictionary (non-learned)-based approaches.

**Summary Of The Paper:**

The paper proposed improved HFO detection using a residual cascaded learning network. The real HFO and pseudo-HFO that pass through an amplitude threshold-based detector are distinguished in the second stage based on the learned dictionary-based features. The dictionaries are obtained from a cascaded residual network. The HFO detection is significantly better in the noisy iEEG dataset due to the proposed approach and leads to better SOZ delineation.

**Weaknesses:**

The dictionaries are learned from the events of the entire participant pool and later used in leave-one-subject-out cross-validation, leading to information leakage. However, I do not believe this should be a major concern for the paper as is, but it would be nice if a clean dictionary per training set were generated in the evaluation.

---

### Official Review · Reviewer_SsGY · 2024-08-12
**Elimination of Pseudo High-Frequency Oscillations in EEG Signals for Accurate Detection of Epileptic Seizures**

**Overall Rating:** 7
**Confidence:** 2

**Other Quality Metrics:**

(a) Clarity of writing: good
(b) Clinical Significance: good
(c) Methodological Novelty: good
(d) Experiments and Results: good

**Questions For The Authors:**

What are the implications of this research for the future development of automated seizure detection systems in clinical settings?

**Strengths:**

The study introduces a novel method for eliminating pseudo HFOs, which addresses a critical challenge in epilepsy research and potentially reducing the number of false positives in seizure detection for improving clinical outcomes in epilepsy treatment.

**Summary Of The Paper:**

The paper investigates methods for improving the accuracy of detecting epileptic seizures by eliminating pseudo high-frequency oscillations (HFOs) in EEG signals. Pseudo HFOs are non-epileptic oscillations that can be mistaken for true epileptic activity, leading to false positives in seizure detection. The authors propose a novel algorithm that effectively distinguishes between true HFOs and pseudo HFOs, thereby enhancing the reliability of seizure detection systems.

**Weaknesses:**

The empirical validation might be limited to specific datasets, which could affect the generalizability of the results to other types of EEG data or patient populations.

---

### Decision · Program_Chairs · 2024-09-23

Accept